# Insights into the Role of microRNAs as Clinical Tools for Diagnosis, Prognosis, and as Therapeutic Targets in Alzheimer’s Disease

**DOI:** 10.3390/ijms25189936

**Published:** 2024-09-14

**Authors:** Nidhi Puranik, Minseok Song

**Affiliations:** Department of Life Sciences, Yeungnam University, Gyeongsan 38541, Republic of Korea; nidhipuranik30@gmail.com

**Keywords:** noncoding RNAs, Alzheimer’s disease, BDNF signaling, biomarker, therapeutic target

## Abstract

Neurodegenerative diseases (NDDs) are a diverse group of neurological disorders characterized by alterations in the structure and function of the central nervous system. Alzheimer’s disease (AD), characterized by impaired memory and cognitive abilities, is the most prevalent type of senile dementia. Loss of synapses, intracellular aggregation of hyperphosphorylated tau protein, and extracellular amyloid-β peptide (Aβ) plaques are the hallmarks of AD. MicroRNAs (miRNAs/miRs) are single-stranded ribonucleic acid (RNA) molecules that bind to the 3′ and 5′ untranslated regions of target genes to cause post-transcriptional gene silencing. The brain expresses over 70% of all experimentally detected miRNAs, and these miRNAs are crucial for synaptic function and particular signals during memory formation. Increasing evidence suggests that miRNAs play a role in AD pathogenesis and we provide an overview of the role of miRNAs in synapse formation, Aβ synthesis, tau protein accumulation, and brain-derived neurotrophic factor-associated AD pathogenesis. We further summarize and discuss the role of miRNAs as potential therapeutic targets and biomarkers for AD detection and differentiation between early- and late-stage AD, based on recent research. In conclusion, altered expression of miRNAs in the brain and peripheral circulation demonstrates their potential as biomarkers and therapeutic targets in AD.

## 1. Introduction

Neurodegenerative diseases (NDDs) represent a significant mental health threat, and are primarily characterized by neuronal loss [1,2]. Despite decades of research, researchers are still searching for potential treatments for these illnesses. Alzheimer’s disease (AD), which is closely linked to aging, is the most prevalent NDD, representing the fifth leading cause of death and the primary cause of dementia globally [3,4]. Currently, approximately 55 million people worldwide suffer from AD, and this number doubles every 5 years [5]. By 2050, approximately 152 million patients are projected to be affected with AD worldwide, with developing nations predicted to experience the largest growth in this number [6]. In affluent nations, over 10% of older (over 65 years of age) individuals have early stage AD, and over 33% of very old (aged > 85 years) individuals may have advanced AD symptoms and indications [7].

Alzheimer’s disease is the most prevalent form of NDD, which progresses to dementia and cognitive deterioration [8]. From a pathological perspective, AD is characterized by the presence of neurofibrillary tangles (NFTs) and plaques of amyloid-beta (Aβ) peptides, leakage of the blood–brain barrier (BBB), neuronal loss, and degeneration [9,10]. According to the amyloid cascade hypothesis, the major proteolytic fragment of amyloid precursor protein (APP) cleavage, Aβ, is the fundamental factor in AD pathogenesis. APP is a single-pass type I membrane protein that has a large extracellular domain and a short cytoplasmic tail, and is found in the somatodendritic and axonal compartments of neurons [11].

Embedded within neurons, misfolded tau protein is the second most important histopathological feature of AD, and is the major cause of tauopathy [12]. The microtubule-associated protein family includes phosphoprotein tau, which is mostly located in axons and, to a lesser extent, in glial cells and somatodendritic compartments. Tau facilitates axonal transit and dendritic organization, in addition to stabilizing and assembling microtubules [13]. Consequently, tau hyperphosphorylation results in the loss of its native function, which ultimately causes deficits in dendritic structure, axonal trafficking, microtube construction, the loss of synapses, neuronal death, and dementia. Prefibrillar aggregates, such as Aβ, are the cause of tau-mediated neurotoxicity [14]. Phosphorylated tau further aggregates to undergo oligomerization before developing into paired helical straight filaments [15].

Although no disease-modifying medication is available to reverse AD, some therapies may lessen the behavioral and cognitive symptoms of patients with AD. The late or final stage of AD is the primary challenge in AD treatment [16,17,18]. Therefore, to improve treatment outcomes, highly sensitive, easy-to-diagnose, and noninvasive peripheral blood-based biomarkers are needed. Cerebral spinal fluid (CSF), whole blood, and plasma are the most appropriate fluids for biomarker detection. To date, both Aβ protein, which produces plaques, and phosphorylated tau, a key component of NFTs, have long been recognized as the two pathological hallmarks of AD [19]; however, both of these have some limitations in terms of their utility in early diagnosis, such as the sensitivity of instruments required to measure these proteins in the blood and CSF, and due to the peripheral production of Aβ, there is significant overlap between diagnostic groups [20].

Recently, numerous studies have examined the relationship between NDDs and noncoding RNAs (ncRNAs) [21,22]. MicroRNAs (miRNAs/miRs) and long noncoding RNAs (lncRNAs) are two of the most important regulatory ncRNAs involved in central nervous system (CNS) homeostasis. Various biological processes, including brain development, maturation, differentiation, neuronal cell specification, neurogenesis, and neurotransmission, are regulated by several types of ncRNAs, and growing evidence has shown that the onset of AD is strongly correlated with miRNAs and lncRNAs [23]. The finding that miRNAs disrupt the expression of the genes implicated in AD has further prompted their application as therapeutic agents [24]. It has also been shown that miRNAs have great promise as therapeutic targets in different regions of the brain; however, further research is needed to fully comprehend the role and mode of action of miRNAs in the CNS [25]. To investigate the functions of miRNAs in AD, researchers have applied both conventional genetic and viral overexpression techniques [26].

Extracellular RNAs (exRNAs) in the CSF are key indicators for early diagnosis [27]. miRNAs are the most extensively researched exRNA species found in circulating fluids, including the CSF, saliva, serum, and urine [28]. Conventionally, Northern blot methods, deoxyribonucleic acid (DNA) microarrays, quantitative reverse transcription polymerase chain reaction (RT-qPCR), and deep sequencing have all been used to quantify and identify miRNAs [29]. However, these methods are time-consuming, labor-intensive, and require expensive supplies and equipment as well as skilled individuals for testing [30]. Nevertheless, they are sophisticated and exhibit good sensitivity and high specificity. Therefore, the development of alternative, affordable, and effective procedures is crucial for the detection and treatment of diseases, including AD [31,32]. A comprehensive evaluation of the current status of biosensor-based methods for amplification-free miRNA detection is also needed. According to current trends, biosensor-based methods for miRNA testing will supplement molecular methods because of their benefits, including amplification-free detection, femtomolar–attomolar LOD, multiplexing ability, and low sample preparation requirements [33].

Novel, noninvasive diagnostic and prognostic biomarkers are all needed to enable the early diagnosis and precision medicine management of AD [34]. Various studies have shown the potential diagnostic, prognostic, and therapeutic roles of miRNAs [35,36], as shown in Figure 1. Hence, in this review, we summarize the involvement of miRNAs in AD pathogenesis, which could serve as potential therapeutic targets as well as diagnostic markers.

## 2. miRNAs: Biogenesis and Function

Noncoding RNAs, also called the “dark matter” of the genome, are RNA sequences that are not translated into proteins; however, recent studies revealed that the majority of lncRNAs have small open reading frames, which means they can encode useful micro-peptides [37]. The previously listed ncRNA families include miRNAs, PIWI-interacting RNAs, lncRNAs, small nuclear RNAs, transfer RNAs, and circular RNAs. Owing to their roles in gene regulation, the lncRNA and miRNA families are among the most therapeutically significant [38]. miRNAs are single-stranded, non-protein-coding, tiny RNAs with a size of approximately 22 nucleotides, that are widely recognized for their ability to bind to the 3′ UTR of messenger RNA and adversely regulate the production of proteins. Approximately 60% of human protein-coding genes are regulated by miRNAs, which are involved in almost all physiological and pathological processes [39,40].

The generation of mature miRNAs is initiated by the transcription of genes by RNA polymerase II in the nucleus. miRNAs are intragenic, processed primarily from introns and very few exons of protein-coding genes. miRNAs are regarded as a family when they are transcribed into clusters, which are defined as single, long transcripts that may share seed regions. miRNA biogenesis can occur via two mechanisms, the canonical and noncanonical pathways. Briefly, primary miRNA is synthesized in the nucleus and catalyzed by the Drosha-DGCR8 complex to form a pre-miRNA, which is transported to the cytoplasm by exportin-5. In the cytoplasm, pre-miRNAs are catalyzed by the Dicer-TRBP (transactivation response element RNA-binding protein)–Argonaute complex to form single-stranded miRNAs, which are also known as mature miRNAs. Mature miRNA and protein combine to create a RISC (RNA-induced silencing complex) and inhibit mRNA translation or degrade the mRNA by binding to a specific sequence at the 3′ UTR of their target mRNAs [41]. 

Small extracellular vesicles (sEVs) represent a novel form of cellular communication. These are composed of proteins and RNAs that are transferred across the distances between cells, potentially causing changes in gene expression and cellular functions [42]. As sEVs can cross the blood–brain barrier (BBB), they are useful tools for molecular diagnostics. Approximately 70% of miRNAs are released from human brain cells into the CNS, with research indicating that they function to control the transcription of almost one-third of all genes. Exosomes, the most researched subtype of EVs, are enriched EVs from which tissue-specific miRNAs may be separated [43,44].

Substantial evidence has linked miRNA dysregulation to NDDs. One of the most exciting new approaches for the treatment of incurable neurological illnesses includes the use of miRNA-based therapies [45]. However, there is a need to create potential pharmacological formulations and delivery systems that can penetrate the BBB and reach the intended brain region, and treat the symptoms. 

The latest approach in miRNA therapeutics in cancer, epilepsy, and others is to either decrease or enhance mRNA expression by targeting miRNAs, primarily based on the following strategies: miRNA mimics (miRNA overexpression), anti-miRNA (miRNA knockdown), and miRNA target blockers [46,47], as shown in Figure 2. Synthetic short double-stranded oligonucleotides that mimic miRNA precursors are known as miRNA mimics. When inserted into cells, the miRNA synthesis machinery identifies and processes them appropriately, allowing them to thereby improve the suppression of the mRNA expression of many targets. Anti-miRNA strands, also known as miRNA inhibitors, block the interaction of miRISCs with target mRNAs or between miRNAs and miRISCs, resulting in mRNA de-repression. Another therapeutic approach is to introduce miRNA-specific target blockers into cells, which results in target-specific mRNA de-repression and boosts mRNA expression [48].

## 3. MiRNA in AD Pathophysiology and as a Therapeutic Target

Numerous studies have previously assessed the functions of miRNAs in animal models of AD. The most popular animal models for AD are APP/PSEN1 transgenic animals, which were created following the finding that familial forms of AD are caused by mutations in the *APP* and *PSEN1* genes [24]. As potential targets for NDD therapy and diagnosis, miRNAs have gained the interest of various researchers. Nonetheless, the outcomes of previous meta-analyses on AD and its correlation with miRNAs have been inconclusive. miRNAs are associated with different pathophysiologies of AD such as tau pathology, Aβ pathology, the BDNF signaling cascade, ER and mitochondrial stress, and synaptic dysfunction, as shown in Figure 3. A summary of preclinical studies supporting the role of miRNAs in AD pathophysiology and their potential therapeutic targets is presented in Table 1. Different miRNAs have different roles and regulatory actions in AD pathophysiology, which are discussed below.

### 3.1. MiRNAs in Synapse Formation

Synapses are believed to be critical in the pathophysiology of numerous neurological disorders, including AD. The roles of synaptosomes and synaptic plasticity in AD have been extensively studied. However, the connections among synaptosomes, miRNAs, and synaptic plasticity in AD and other neurological disorders are not well understood [105]. Maintaining proper synaptic and cognitive functioning as well as neurotransmission depends on structurally and functionally active synapses. One of the primary physiological characteristics of AD is synaptic loss and destruction, which also plays a role in the development of behavioral and physiological traits [106,107].

Synaptic miRNAs are found in small, microscopic neuronal compartments, such as synaptosomes, synaptic vesicles, and neural dendrites. Synaptic miRNAs regulate multiple synaptic processes that result in normal brain function and activity. However, the exact function of synaptic miRNAs in AD progression remains unclear. The existence of miRs at the synapse, in synaptic compartments, and their involvement in many synaptic processes are highlighted in this review. Kumar S and Reddy PH previously highlighted the change of synaptic miRs in AD and the impact of miRNA regulation on synaptic functioning, and further discussed how synaptic miRNAs affect synaptic adenosine triphosphate synthesis, mitochondrial function, and synaptic activity as AD progresses [105].

Synaptic function deficiencies may play a role in the preclinical stage of AD development. According to a recent study examining the expression levels of particular miRNAs linked to synaptic proteins, plasma from patients with mild cognitive impairment (MCI) and AD showed dysregulated levels of miR-92a-3p, miR-181c-5p, and miR-210-3p, whereas no alterations were observed in the plasma of patients with FTD. Furthermore, the combination of these three miRNAs produced a high diagnostic accuracy in differentiating between people with AD and MCI from healthy controls and may also predict the progression of MCI to AD. Therefore, the molecular signatures of miR-92a-3p, miR-181c-5p, and miR-210-3p in plasma can be used as a new biomarker for AD [108].

A molecular connection between miR-502-3p and GABAergic synapse activity in an AD mouse hippocampal neuronal cell line was examined by Sharma B et al., 2024. The results of the investigation showed that while miR-502-3p suppression raised the amount of the GABRα1 protein, miR-502-3p overexpression decreased the level of GABRα1. Overall GABA function was discovered to be enhanced as a result of miR-502-3p overexpression, indicating a negative association between miR-502-3p levels and GABAergic synapse function. The whole research suggested that miRNA, namely miR-502-3p, may be a viable therapeutic target for altering the function of GABAergic synapses in AD [109]. 

Rodriguez-Ortiz CJ and colleagues found that miR-181a is a major negative regulator of the cellular events required for synaptic plasticity and memory and leads to synaptic dysfunction and memory impairments in AD [110]. 

A recent study by Ge J and their team, 2023 [111], revealed that overexpression of miR-431 in the CA1 region of the hippocampus in AD mice improved their memory deficits and synaptic plasticity but did not affect Aβ levels. miR-431 was identified as targeting Smad4, and knocking down Smad4 mitigated synaptic plasticity and memory impairments in APP/PS1 mice by influencing the expression of synaptic proteins such as SAP102. Additionally, the protective effects of miR-431 were negated by Smad4 overexpression, indicating that miR-431 alleviates synaptic damage partly through suppressing Smad4. These results suggest that targeting the miR-431/Smad4 pathway could be a promising approach for AD treatment.

Kumar S and colleagues (2022) investigated changes in cytosolic and synaptosomal miRNAs in postmortem brains of AD patients, healthy brains, Tg2576 mice, tau transgenic mice, and wild-type mice. The study pinpointed synaptosomal miRNAs that were upregulated with AD progression and highlighted three particularly promising miRNAs—miR-501-3p, miR-502-3p, and miR-877-5p—that influence the function of excitatory and inhibitory synapses in AD [112].

### 3.2. MiRNAs and Brain-Derived Neurotrophic Factor (BDNF) Signaling

Brain-derived neurotrophic factor is a growth factor belonging to the neurotrophin family which is essential for nervous system development. It also promotes neurogenesis and helps existing neurons survive. Abnormal BDNF levels in the blood and brain are linked to the etiology of NDDs, including AD [113]. By binding to the tropomyosin receptor kinase B (TrkB) receptor, BDNF initiates downstream signaling cascades that result in cell growth, multiplication, survival, and neurogenesis. Given the significance of BDNF function, determining the regulatory mechanisms governing BDNF expression levels may open the door to developing more precise and effective NDD treatments.

Recently, miRNAs have been suggested as potential novel therapeutic targets for the treatment of diverse CNS illnesses. However, there are significant obstacles in translating these discoveries into useful anti-miRNA treatments, particularly when the targets remain within the central nervous system [114]. 

miRNAs are important regulators as they specifically target the 3′ untranslated regions of genes. According to a recent study by Sun et al. (2022) [115], approximately 127 miRNAs may have a regulatory role in BDNF expression. Brains, blood, and CSF from patients with AD show upregulation or downregulation of a broad spectrum of miRs. Twenty-seven differentially dysregulated miRNAs involved in amyloidogenesis, inflammation, tau phosphorylation, apoptosis, synaptogenesis, neurotrophy, neuron disintegration, and the activation of cell cycle entry have thus far been identified.

Wu BW and colleagues found that miR-10a overexpression inhibits synapse remodeling in hippocampal neurons and neuronal cell proliferation and apoptosis through the BDNF-TrkB signaling pathway in a rat model of AD. 

Wei Li and their team’s research verified that miRNA-613 controls BDNF expression in SH-SY5Y cells and APP/PS1 transgenic mice. The study first hypothesized that miR-613 targets the 3′-UTR of BDNF and experimentally confirmed that BDNF mRNA and protein expression levels were markedly decreased in the presence of miR-613; conversely, BDNF expression levels were enhanced when miR-613 was suppressed. This result implies that, in the AD model, miR-613 negatively regulates BDNF expression [116].

Zhang J et al., 2018 studied the role of miR-322 (a homolog of human miR-424) in the Tg2576 mouse model and found that miR-322 stimulates tau phosphorylation by inhibiting the activation of the BDNF–TrkB receptor; in the absence of miR-322, tau phosphorylation is attenuated and TrkB activity is restored. Concludingly, their research revealed a unique miRNA-dependent mechanism of BDNF degradation in AD pathogenesis, which may motivate therapeutic approaches based on miRNAs or BDNF.

### 3.3. miRNAs and Aβ Protein

Two prominent pathologies associated with AD include excessive Aβ synthesis, primarily caused by the APP cleaving enzyme 1 BACE1, and tau hyperphosphorylation, which results from an imbalance in the kinase/phosphatase system and the activation of protein kinase A (PKA) [117]. Growing data suggest that elevated APP expression may stimulate the synthesis of Aβ, which could drive neurotoxicity, synaptic dysfunction, and ultimately dementia. Numerous miRNAs are involved in controlling the expression of APP [118]. 

The function and underlying molecular mechanisms of miR-200a-3p in mediating neuroprotection against AD-related impairments were studied by Wang L and colleagues [101]; miR-200a-3p expression was assessed in the hippocampal regions of APP/PS1 and SAMP8 mice, an in vitro model of AD, and in blood plasma from patients with AD. Using dual luciferase assays and bioinformatics analysis, the targets of miR-200a-3p were identified. The microarray miRNA profile revealed decreased levels of miR-200a-3p, which reduced cell death and inactivated the Bax/caspase-3 axis. Additionally, it downregulated levels of tau phosphorylation and Aβ1-42 in vitro. Reduced Aβ1-42 generation and tau hyperphosphorylation were observed simultaneously as a result of controlling BACE1 protein translocation and the protein kinase cAMP-activated catalytic subunit beta linked to the three primary UTRs. Overexpression of PRKACB or BACE1 in cultured cells was further found to disrupt the action of miR-200a-3p. This study’s findings indicate that miR-200a-3p may have a role in the pathophysiology of AD and protect neurons from Aβ-induced toxicity through two potential mechanisms: one involves preventing Aβ overproduction by reducing BACE1 expression, and the other involves reducing tau hyperphosphorylation by reducing PKA expression.

Kumar et al. (2019) [74] investigated how miR-455-3p protects against AD pathogenesis by regulating APP, synaptic modulation, mitochondrial biogenesis and dynamics, cell survival, and apoptosis. The immunological results demonstrated that miR-455-3p decreased the amounts of mutant APP, Aβ(1-40) and Aβ(1-42), and C99, and confirmed that miR-455-3p bound to the 3′UTR of the APP gene. Increased levels of mitochondrial biogenesis and synaptic gene mRNA and protein were concurrently observed with the increased expression of miR-455-3p. Furthermore, miR-455-3p increased the expression of fusion proteins (OPA1, Mfn1, and Mfn2), and decreased the expression of mitochondrial fission proteins (DRP1 and FIS1). These findings led us to conclude that miR-455-3p controls APP processing and protects against mitochondrial and synaptic defects caused by mutant APP in AD [74].

Zhu HC and colleagues [119] previously uncovered the function of miR-195 in the production of Aβ. A negative correlation was also found between the expression levels of miR-195 and the BACE1 protein in SAMP8 mice, which was explained by the prediction of miR-195 binding sites within the BACE1 3′-UTR using bioinformatics algorithms. A luciferase assay allowed the determination of the location of the target in HEK293 cells. In N2a/WT cells, BACE1 protein levels were reduced by the overexpression of miR-195; however, the level of the BACE1 protein was enhanced by the suppression of miR-195. Moreover, Aβ levels were lowered by miR-195 overexpression in N2a/APP but Aβ levels were raised by miR-195 inhibition. Thus, miR-195 could suppress BACE1 translation to downregulate the amount of Aβ, the overall conclusion being that miR-195 may represent a new target for AD treatment [119].

miR-346 contributes to the overexpression of APP in the CNS and helps preserve APP control of iron, which is disturbed in the later stages of AD. In HeLa cells, miR-346 significantly increased the expression of both the natural APP protein and an APP 5′-UTR reporter clone. These effects were shown to be mediated by a specific contact with the predicted APP 5′-UTR target region by site mutagenesis and TP transfections. In human primary cultures, an upregulatory effect was observed; however, it was only observed after iron chelation. As a result, through a “noncanonical” target site in the APP 5′-UTR that also contains an IRE, miR-346 has “noncanonical” regulatory effects on APP expression. One possible treatment approach to control APP expression and Aβ synthesis is to inhibit the observed connection [120].

miR-101 controls APP levels in human cell cultures of various ancestries; lowering APP expression has been proposed as a potential tactic for lowering the degenerative processes underlying AD since APP and Aβ are key components of the pathways linked to the disease. These findings suggest that miR-101 is a novel therapeutic target for APP-level regulation. MiR-101 target sites reside within the APP 3′-UTR and an experimental study confirmed that miR-101 significantly reduced the expression of APP 3′-UTR [121]. On chromosomes 1 and 9, two distinct genomic loci, one situated in an intronic region of RCL1 and the other in an intergenic region, are responsible for the expression of MiR-101. The miR-101 promoter elements that control transcription are poorly defined and are only currently being identified. Understanding the regulatory components that govern the expression of miR-101 could unveil innovative approaches to regulate its expression, and therefore the expression of APP. Subsequent investigations are required to analyze this regulatory network, and to ascertain whether it could be modified in a manner that could be beneficial as a possible treatment for AD [121].

### 3.4. miRNAs and Tau Protein

Tau is the most frequently expressed microtubule-associated protein in the neurons of the human brain. Tang M’s (2023) study confirmed the potential role of miRNAs in tau production regulation. miRs such as miR-125b and miR-219-5p can control tau phosphorylation by targeting tau-related enzymes, including GSK-3ß and CDK5/P35/25. miR-322 is also crucial for controlling BDNF, which in turn increases tau hyperphosphorylation. Furthermore, increased miR-138 and miR146a levels cause an increase in tau hyperphosphorylation. Retinoic acid receptor alpha and rho-associated coiled-coil-containing protein kinase 1 (ROCK1) are the signaling pathways primarily targeted by these two miRNAs. Tau mRNA is regulated by miR-132, whereas tau protein toxicity can be either up- or downregulated by miR-124-3p, which also regulates CAPN1 (Calpain 1) mRNA. miR-106b targets Fyn, which in turn controls the production of Aß42, hence preventing tau hyperphosphorylation. Similarly, miR-128 and miR-137 both suppressed the NF-kB pathway and CACNA1C expression. By examining variations in their levels, several alternating miRNAs that target other genes to control tau proteins may be analyzed to provide a broad notion of how miRNAs might be employed as diagnostic biomarkers [122].

Research has further shown that hsa-miR-143-3p inhibits translation and directly reduces the amount of the DAPK1 (death-associated protein kinase 1) protein. Furthermore, miR-143-3p promotes neurite outgrowth and microtubule assembly, whereas hsa-miR-143-3p mimics decreased the amount of phosphorylated tau at several AD-related locations. Furthermore, miR-143-3p reduces the production of Aβ and APP phosphorylation. By targeting DAPK1, miR-143-3p’s capacity to reduce tau phosphorylation, APP phosphorylation, and Aβ production was countered by restoring DAPK1. Additionally, in the hippocampal tissues of patients with AD, hsa-miR-143-3p levels were found to be reduced and inversely linked with DAPK1 expression. This finding points to the possible functions of hsa-miR-143-3p in regulating the proteins associated with AD, such as tau dysfunction and Aβ accumulation. Although miR-143-3p facilitates microtubule assembly and neurite outgrowth, it reduces tau phosphorylation. Additionally, miR-143-3p decreased the production of Aβ 40 and Aβ 42 and inhibited APP phosphorylation [123].

According to Smith PY and colleagues [117], various miRNAs have regulatory roles in BDNF expression. Furthermore, bioinformatics analysis also revealed the top ten roles of frequently occurring miRNAs in the candidate studies. Commonly downregulated miRNAs primarily target DNA-templated transcription, whereas commonly upregulated miRNAs largely target the nucleus. Cooperation in miRNA signatures is revealed by a thorough examination of all miRNA investigations, whether in brain tissue, CSF, or peripheral blood. Early preclinical phase identification offers a critical opportunity for therapeutic interventions.

Similarly, in P301S tau transgenic mice, miR-132 was found to improve long-term potentiation while reducing PHF-tau pathology and neurodegeneration. Direct control of the tau modifiers acetyltransferase EP300, kinase GSK3β, RNA-binding protein Rbfox1, and proteases Calpain 2 and Caspases 3/7 mediates the neuroprotective effects [124]. An analysis of miRNA expression patterns in patients with PSP (progressive supranuclear palsy) revealed that miR-132 is specifically downregulated in sporadic PSP, a prominent 4R-tau tauopathy. This study showed that the neuronal splicing factor PTBP2, whose levels are elevated in patients with PSP, is a direct target of miR-132. Endogenous 4R/3R-tau ratios in neuronal cells were similarly affected by miR-132 overexpression or PTBP2 knockdown. Lastly, this study presents evidence that, at the period during 4R-tau expression throughout postnatal brain development, miR-132 and PTBP2 have an inverse correlation. Together, these findings point to the potential function of miRNAs in a subset of tauopathies, and indicate that modifications in the miR-132/PTBP2 pathway may be responsible for the aberrant splicing of tau exon 10 in the brain [125].

Strathmann J et al. (2014) [126] studied the role of miR-125b in AD pathogenesis and found that the overexpression of miR-125b resulted in tau hyperphosphorylation and the elevation of p35, cdk5, and p44/42-MAPK signaling in primary neurons. As direct targets of miR-125b, the phosphatases DUSP6 and PPP1CA, as well as the anti-apoptotic protein Bcl-W, were downregulated. Tau hyperphosphorylation can be induced by knocking down these phosphatases, while tau phosphorylation induced by miR-125b is prevented by the overexpression of PPP1CA and Bcl-W, indicating that these phosphatases mediate the effects of miR-125b on tau. In contrast, tau phosphorylation and kinase expression/activity were decreased when harsh decoys restricted miR-125b expression in neurons. Mouse hippocampi injected with miR-125b decreased associative learning and increased tau phosphorylation in vivo by downregulating the expression of Bcl-W, DUSP6, and PPP1CA. Notably, AD brains exhibited decreased levels of DUSP6 and PPP1CA. These studies confirm the role of miR-125b in AD pathogenesis by promoting tau phosphorylation.

MiRNA-146a was found to be upregulated in AD patient brains and miR-146a suppressed ROCK1 activity. ROCK1 binding to PTEN (phosphatase and TENsin homolog deleted on chromosome 10) is necessary for PTEN phosphorylation, which promotes tau dephosphorylation. AD is also associated with the reduction in PTEN phosphorylation of PTEN and PTEN-immunoreactive temporal lobe pyramidal neurons. The ROCK1-PTEN signaling pathway in neurons may be regulated by miR-146a upregulation, which could lead to aberrant tau hyperphosphorylation in neurons. The results of Wang G et al. 2016 study provide credence to the theory that an overexpression of dysregulated microRNA-146a in neurons negatively controls the translation of the ROCK1 protein. Decreased neural PTEN phosphorylation results from a decrease in the neuronal ROCK1 protein, which impairs neuronal tau dephosphorylation [127].

According to Walgrave H et al. (2023), miR-132 is one of the few miRNAs that is reliably and strongly downregulated in the brains of patients with AD. Memory impairment and adult hippocampal neurogenesis are restored and amyloid and tau pathologies are reduced in AD mouse brains by miR-132 upregulation. However, before miR-132 supplementation can be used in AD therapy, a thorough examination of the consequences of protein functional pleiotropy is necessary [128].

PTEN has also been extensively studied in AD-related postsynaptic dysfunction. ER stress increases miR-200c expression in early-stage AD. Wu Q and colleagues investigated the mechanism by which the miR-200 family regulates PTEN expression in neural cells. They discovered that Aβ deposition causes neuronal ER stress, which in turn triggers miR200c expression in a mouse model of AD caused by APPs and PS1E9 double transgenes. In AD brains, pharmacological inhibition of ER stress reduced the overexpression of miR-200c caused by Aβ, and human AD patients as well as AD mice serums contained miR-200c. This suggests that miR-200c is a component of the intrinsic adaptive machinery of neurons and played a protective role through PTEN suppression [129].

Liu et al. [130] also showed that in a double-transgenic mouse model of AD, the expression levels of miR-26b were increased, additionally finding that while the suppression of miR-26b in N2a/APP cells increased IGF- 1 protein levels and inhibited the synthesis of Aβ, the overexpression of miR-26b in N2a/APP cells downregulated IGF-1 protein expression levels and encouraged the creation of Aβ. Furthermore, a luciferase assay in HEK293 cells was used to corroborate the miR-26b target locations in IGF-1. Increasing IGF-1 expression levels has been proposed as a potential therapeutic approach for AD. The findings of this investigation indicate that miR-26b may be a useful therapeutic target for the inhibition of Aβ production.

## 4. miRNA as an Alzheimer’s Disease Biomarker

The primary biomarkers of AD include the tau protein and Aβ peptide levels in body fluids. However, these methods have certain drawbacks, and the discovery of new biomarkers has made it possible to seek quicker and more precise diagnoses. As discussed in Section 3, miRNAs are involved in the pathophysiology of AD, and patients with neurodegenerative diseases, such as AD, have significantly different amounts of several miRNAs compared to healthy controls [131], indicating that miRNAs may be useful as diagnostic and prognostic biomarkers. miRNAs can be measured easily and noninvasively, and are stable in bodily fluids such as blood and CSF [122]. Various in vitro and in vivo studies demonstrating the potential role of miRNAs as biomarkers for AD diagnosis are listed in Table 2.

Yoon et al. recently conducted a meta-analysis of 334 miRNAs associated with AD. Of these, 56-miRs were found to upregulated or downregulated in AD and MCI patients compared with healthy controls. The study found that the upregulation of miRNA is mainly associated with pathways linked to the immune response, inflammation, and apoptosis [150]. Similarly, to find original research comparing the differential expression of miRNAs in AD cases against controls, Takousis P et al. conducted a systematic review. Following a comprehensive assessment, data from 147 separate data sets reporting information on 1121 distinct miRNAs were combined from 107 qualified research studies. A total of 57 miRNAs were discovered to be differentially expressed at study-wide significance across 461 meta-analyses. It was observed that several miRs had differential expression in the brain, CSF, and blood-derived specimens [151]. 

Ines Maldonado-Lasuncion IM and colleagues investigated the serum expression of AD-related miRNAs and discovered that miR-455-3p, miR-34a-5p, and miR-146a-5p are strong biomarkers for AD, as patients with AD have notably elevated levels of these miRNAs. Conversely, postmortem brain tissues had higher levels of miR-34a-5p and miR-146a-5p, but the plasma and CSF of patients with AD had lower levels of these molecules [152]. Kumar et al. (2019) [153] found that miR-455-3p is consistently upregulated in the CNS, serum, and plasma of patients with AD. Moreover, the plasma levels positively correlated with Aβ1-42 levels in CSF. A recent study also examined the relationship between the blood levels of particular miRNAs (miRNAs-9-5p, 29b-3p, 146a-5p, and 125b-5p) linked to AD and alterations in cognition and cerebral cortex integrity [154]. 

Numerous studies have consistently demonstrated the downregulation of miR-9-3p and miR-29b-3p in the brain and blood samples of AD patients, whereas miR-125b-5p has been reported to be upregulated in the brain and cerebrospinal fluid and downregulated in the serum and plasma. Remarkably, reduced levels of serum expression of miR-9-3p, miR-34a-5p, and miR-125b-5p are linked to alterations in cognitive function. In healthy older individuals, low levels of miR-9-3p, miR-29b-3p, miR-34a-5p, and miR-125b-5p are associated with cortical thickness, while low levels of miR-29b-3p, miR-125b-5p, and miR-146a-5p are associated with aberrant cortical glucose metabolism [154]. Brain-enriched miRNAs, including miR-9-5p/3p, control development, and alterations in these miRNAs have been linked to many neurological conditions, including AD. miR-9-3p is essential for memory and synaptic plasticity in adult mice [155].

Remarkably, a recent study on the expression of miRs previously linked to inflammation found that, when compared to matched controls, the plasma levels of miR-17-5p, miR-21-5p, and miR-126-3p were significantly higher in patients with AD, and that the cognitive impairment experienced by patients was inversely correlated with both miR-21-5p and miR-126-3p [156]. 

Wiedrick et al. (2019) [157] conducted validation research to determine the validity of miRs as a biomarker for AD in human CSF and discovered that miRNA expression in CSF from living donors can differentiate patients with AD from controls. Of the 37 miRNAs identified in the discovery study, 26 were still able to distinguish patients with AD from controls. A strong set of biomarkers comprising the miRNAs verified in this study will now be further assessed for potential applications as clinical biomarkers for AD.

Kumar et al. [158] reported the discovery and validation of a unique circulating 7-miRNA signature (hsa-let-7d-5p, hsa-let-7g-5p, hsa-miR-15b-5p, hsa-miR-142-3p, hsamiR-191-5p, hsa-miR-301a-3p, and hsa-miR-545-3p) in plasma, which could distinguish patients with AD from healthy controls with 95% accuracy. All signature miRNAs showed a 0.2-fold difference between the AD and NC samples, with *p*-values of 0.05. Pathway analysis, considering the enriched target mRNAs for these signature miRNAs, was also conducted, suggesting that the disruption of multiple enzymatic pathways, including lipid metabolism, could be a factor in the etiology of AD.

According to Liu CG et al. (2021), miR-106b is a diagnostic biomarker for AD which has predictive value for the response to rivastigmine treatment. They used real-time PCR to determine miR-106b serum concentrations and found that serum samples from patients with AD had considerably lower levels of miR-106b than those from controls. Receiver operating characteristic (ROC) analysis revealed 62% specificity and 94% sensitivity. The study concludes that serum miR-106b is a potential serum biomarker for the early diagnosis of AD [132].

Prior research has also shown that BACE1 dysregulation may be a critical factor in AD pathophysiology. β-site amyloid precursor protein cleaving enzyme 1 (BACE1) protein is an endopeptidase that cleaves the Aβ precursor protein to create neurotoxic Aβ peptide Aβ 1–42. Among the miRNA expression changes associated with AD, miR-107 was unique in that its levels declined markedly, even in patients exhibiting early stages of the disease. Reduced neuronal miR-107 expression has been observed in specific cerebral cortical laminas implicated in AD pathology using hybridization with cross-comparisons with neuropathology. BACE1 post-transcriptional regulation is influenced by miR-107, and this pathway may aggravate disease in AD brains [159].

Studies in both AD patients and cell models showed that miR-149 directly binds the 3′-UTR of BACE1 and has a negative correlation with BACE1. ROC analysis further suggested that reduced serum miR-149 levels had a comparatively high diagnostic accuracy in separating AD patients with AD from healthy controls. Furthermore, compared to PD patients, AD patients had significantly lower serum miR-149 levels, which could accurately differentiate AD from PD cases. Serum miR-149 levels can differentiate between patients with severe AD and those with mild-to-moderate AD, suggesting that serum miR-149 may be a viable biomarker for AD diagnosis and predict the severity of AD [160].

miR-384 is expected to target the 3′-UTRs of BACE-1 and APP. In one study, Liu et al. [161] discovered that APP and BACE-1’s mRNA and protein expression were both decreased by miR-384 overexpression. miR-384 inhibitors trigger the overexpression of BACE-1 and APP, in turn influencing protein activity. Using the luciferase assay, the binding sites of miR-384 on the 3′-UTRs of APP and BACE-1 were identified. Moreover, experiments showed that the treatment of cells with Aβ42 suppressed the expression of miR-384, which resulted in a persistent decline in miR-384 expression.

Barros-Viegas AT et al. [73] studied the role of miRNA-31 in improving cognition and abolishing Aβ pathology by targeting APP and BACE1 in an AD mice model, discovering that miR-31 was downregulated in patients with AD, and may concurrently lower the levels of APP and BACE1 mRNA in the hippocampal regions of AD mice. This resulted in a notable amelioration of memory impairment, as well as a decrease in anxiety and cognitive rigidity. Furthermore, in this paradigm, lentiviral-mediated miR-31 expression markedly improved AD neuropathology by substantially lowering Aβ accumulation in the hippocampus and subiculum. Moreover, the buildup of glutamate vesicles in the hippocampal region decreased to levels observed in age-matched nontransgenic animals, with a corresponding increase in miR-31 levels. Overall, the findings of this study suggest the possibility of using miR-31-mediated APP and BACE1 regulation as a therapeutic approach for the treatment of AD.

Wang L et al. [101] further found that the pathology of AD is associated with miR-200a-3p, which exerts neuroprotective effects against Aβ-induced toxicity through two possible mechanisms: one involves inhibiting Aβ overproduction by suppressing BACE1 expression, and the other involves attenuating PKA expression to reduce tau hyperphosphorylation. In this study, the authors examined the expression of miR-200a-3p in the hippocampus of APP/PS1 and SAMP8 mice and blood plasma from patients with AD. Microarray miRNA profile studies indicated that miR-200a-3p is downregulated, suggesting that miR-200a-3p may be a possible biomarker of AD.

The miRNAs miR-135a, -200b, and -429, which exhibit notable changes in AD and potentially target APP or BACE-1, were examined by Liu et al. in 2024. Using qPCR, the dysregulation of these miRNAs was further validated in samples from patients with AD and animals. Researchers have discovered that miR-135a is directly associated with the 3′-UTR of BACE-1 and suppressed its production and activity. However, the 3′-UTR of APP was the target of miR-200b and -429, which were considerably downregulated in APP/PS1 transgenic mouse hippocampi when compared to those of the wild-type control. The serum levels of miR-135a and -200b were notably lower than for those in the control group. The MCI group had a higher serum miR-200b level than the Alzheimer’s type dementia (DAT) group but a lower level than the control group. Taken together, these results indicate that miR-135a, -200b, and -429 may be involved in the development of AD; miR-200b shows particular promise as a noninvasive, readily identifiable blood-based biomarker for patients with MCI and DAT [162].

## 5. Challenges and Future Prospective

Unlike traditional drugs, miRNAs work at a molecular level and control target protein expression by directly binding to sequences in the mRNA of relevant signaling pathways. However, for miRNAs to function at high doses, they must be systematically delivered to the CNS, and crossing the BBB is challenging. Although local brain drug delivery is challenging to achieve using current treatments, it has considerable effects on animal therapy research. Local drug delivery is anticipated to be utilized in the future to distribute miRNAs to the brain to treat AD, provided that clinical drug delivery technologies continue to advance [163].

Advances in pharmaceutics are required to improve the potency and efficiency of drugs, optimize their properties, and ensure their safety. Advances in the chemical and material sciences have enhanced the administration, delivery, and modification of medications. To some extent, when creating miRNA mimics and anti-miRNAs, one may consult guidelines for creating medications based on siRNAs and ASO, respectively. Preclinical research requires the use of preconditioned animal models. Furthermore, to determine the best method to administer a drug and assess its toxicity in the human body, thorough clinical trials are required. Though some clinical trials have been carried out, no outcomes have been obtained as of yet. This could be because of small sample sizes, the intrusive character of the testing procedures, and the general public’s difficulty accepting and analyzing the results. To confirm that miRNAs are biomarkers for identifying neurodegenerative illnesses, a few clinical trials are now being prepared for enrollment and investigation [164].

The delivery processes, including BBB permeability, particle instability, low trafficking to the target sites, biodistribution within tissues, and improper cargo release from delivery vehicles, represent the largest obstacles to the development of effective miR-based therapies. The development of efficient nanocarriers for the delivery of miRNAs may be aided by the development of nanomedicines. Furthermore, to provide accurate physicochemical characterization and assessment of the in vivo activity of a drug, batch-to-batch variability must be eliminated during the development of drug systems by carefully adjusting the nanoparticle composition and miRNA loading [165]. Therefore, to provide a unified approach for the manufacturing and assessment of medications, suitable regulatory concerns and quality control techniques are required.

From a biological perspective, there is growing evidence that miRNAs are crucial for regulating many biological processes, such as the formation and normal operation of the central nervous system, and miRNAs are shown to reflect the degenerative process taking place in the brain tissue and are differentially expressed in patients compared to controls [166]. The analysis of miRNAs as a potent diagnostic biomarker faces numerous pre-analytical and analytical obstacles. However, standardizing methods and analyses is necessary to bring them into clinical practice and ensure low costs as well as high specificity and sensitivity. Another challenge is finding certain miRNA patterns that can discriminate between different conditions such as separating AD patients from healthy people, determining the disease’s severity, and separating AD patients from those with neurological disorders. However, when a sufficiently large panel of miRNAs is analyzed, distinct patterns of miRNAs may still be found concerning certain disorders, even despite these overlaps. This may improve the utility and specificity of miRNAs as biomarkers. The absence of reference level, analytical standardization, and clinical validation remain significant issues with miRNAs as a biomarker.

## 6. Conclusions

Biofluid-derived indicators are currently being investigated as potential biomarkers for the early and preclinical diagnosis of indicators of AD. Many miRNAs that regulate AD-associated proteins in the brain have been reported to be dysregulated. These molecules target gene networks in the brain that play crucial roles in synapse formation and physiology. Therefore, miRs may be promising therapeutic targets for the treatment of AD, and measuring miRNA levels in peripheral blood may help distinguish between individuals with late-stage AD and those whose levels change in the early stages of the disease. Therefore, miRNAs may be used as potential biomarkers for the early prognosis of AD.

## Figures and Tables

**Figure 1 ijms-25-09936-f001:**
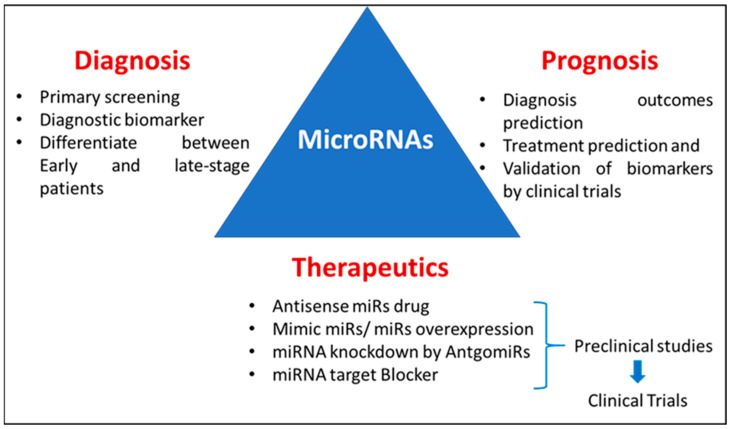
Diagrammatic representation of a possible role of miRNAs in diagnosis, prognosis, and therapeutics.

**Figure 2 ijms-25-09936-f002:**
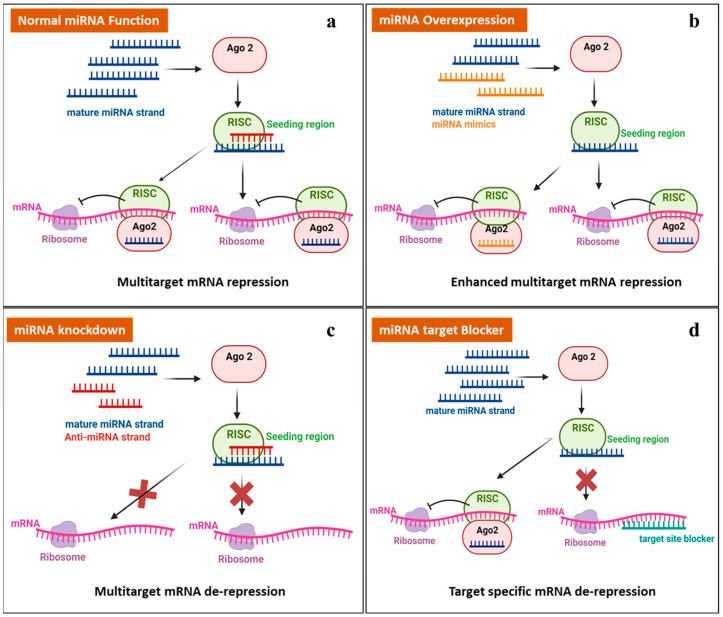
Therapeutic approaches for manipulating miRNA. (**a**) The normal mode of action of miRNAs. (**b**) MiRNA mimics: an increased level of miRNA leads to the downregulation of target mRNA expression. (**c**) Anti-miRNA: miRNA knockdown leads to the overexpression of target mRNA. (**d**) Introducing a target-specific miRNA target blocker in a cell that leads to the overexpression of target mRNA. (Created with biorender.com, accessed on 7 May 2024).

**Figure 3 ijms-25-09936-f003:**
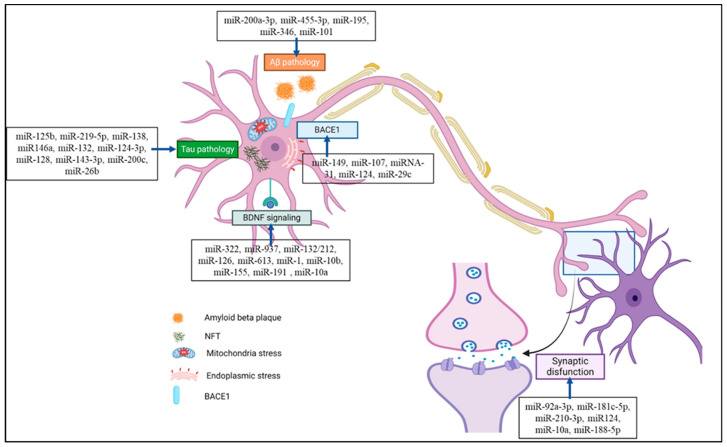
miRNA is associated with various pathophysiologies of Alzheimer’s disease, including tau-, Aβ-, BACE1-, and BDNF signaling-associated pathologies, and is also related to synaptic dysfunction. miRNA associated with AD pathology could be a potential therapeutic target or biomarker for its diagnosis. (Created with BioRender and accessed on 9 September 2024).

**Table 1 ijms-25-09936-t001:** A summary of preclinical studies supporting the role of miRNA in AD pathophysiology and therapeutics.

miRNA	Experimental Model	Upregulated/ Downregulated	Outcome	Parameter Study and Technique Used	Concluding Remark	Reference
miR-21–5p	hNSCs	Increased	Reduced neuronal autophagy by focusing on the suppression of cytokine signaling and RAB11A	RT-qPCR and Western blot	Therapeutic target	[49]
miRNA-511-3p	AD patient serum samples	Downregulated	Aβ1-40-triggered cell viability and inflammation were regulated by miR-511-3p	RT-qPCR and Western blot	Candidate biomarker in AD	[50]
miR-146a-5p	APP/PS1 mice	Mir-146a-5p level increase in hippocampus	Modifies mature hippocampal neurogenesis abnormalities in APP/PS1 mice using Klf4/p-Stat3 signaling	RT-qPCR and Western blot	Therapeutic target	[51]
miRNA-142–5p	Aβ1–42 oligomer-injected rats	Overexpressed	Enhances memory and learning in rats through regulating the PTPN1-mediated Akt pathway	miRNA expression level (RT-qPCR and Western blot assessments) and behavior assay (Morris water maze (MWM) and NOR)	Therapeutic target	[52]
hsa-miR-365b-5p	Autopsy sample	Upregulated	Regulates neuroinflammation via Wnt and the oxidative stress pathway	Microarray bioinformatics analysis	Key therapeutic target	[53]
miR-34c	SAMP8 mice	Upregulated	Through the ROS-JNK-p53 pathway, increased miR-34c targets synaptotagmin 1 to mediate synaptic impairments in AD	Dual luciferase assay	Promising novel therapeutic target	[54]
miR-532-5p	5XFAD mice	Downregulated in AD	Rescues BBB function via the downregulation of EPHA4	miRNA expression level (RT-qPCR, Western blot, and in situ hybridization) and behavior assay (MWM test and eight-arm radial maze)	Potential therapeutic target	[55]
miRNA-126a-3p	Male APP/PS1 mice	Downregulated in AD	Participates in hippocampal memory via AD-related proteins	miRNA expression level (RT-qPCR and Western blot assessments) and behavior assay (MWM test and NOR)	Therapeutic target for AD memory disorders	[56]
miR-342-5p	APP mice (Exo-APP)	Upregulated/downregulated in AD	Ameliorates Aβ formation via targeting BACE1 in experimental mice	miRNA expression level (RT-qPCR, Western blot, and dual luciferase reporter assay)	Potential miRNAs in AD clinical therapy	[57]
miRNA-128	5XFAD mice	Downregulated	Decreases the accumulation of Aβ and lowers tau phosphorylation by blocking the expression of GSK3β, APPBP2, and mTOR	miRNA expression level (luciferase reporter assay) and behavior assay	Promising molecular target	[58]
miR-146a-5p	Aβ_1-42-injected mice; miR 146a-5p and Aβ1-42-treated mice; and miR-146a-5p inhibitor- and Aβ1-42-treated mice	NA	Exacerbates AD-like disease and cognitive impairment by activating oxidative stress via MAPK signaling	miRNA expression level (RT-qPCR and Western blot) and behavior assay (MWM test)	Molecular target	[59]
miR-16-5p	5xFAD	Upregulated	miR-16-5p targets the B cell, and an anti-apoptotic factor	Microarray and RT-qPCR analysis	Therapeutic target	[60]
miR-135a-5p	APP/PS1 mice	Downregulated	Through the Rock2/Add1 signaling pathway, miR promotes memory impairments	Luciferase reporter assay, RT-qPCR, and behavior assay (MWM test and NOR)	Therapeutic strategy for AD	[61]
miR-146a	B6/JNju-Tg (APP/PS1) mice	Abnormal	Overexpression of the target gene negatively affects cognitive behaviour—learning and memory	miRNA expression level (RT-qPCR) and behavior assay (MWM test and NOR)	Therapeutic target	[62]
miR-212 and miR-132	Alzheimer’s patient CSF and blood samples	Downregulated	Decreased levels of both miR-132 and miR-212 in CSF; however, miR-132 was also downregulated in blood	RT-qPCR	Potential theragnostic	[63]
MiR-29c-3p	Male SPF C57BL/6J mice	Upregulated	Since miR-29c-3p binds specifically to the 3′-UTR of BACE1, it adversely affects BACE1 levels, which in turn slows the advancement of AD	miRNA expression level (RT-qPCR and Western blot assessments)	Therapeutic target	[64]
miR-206-3p	SPF C57 mice	Upregulated	BDNF expression enriches neuronal morphology and recovers cognitive ability and memory	miRNA expression level (RT-qPCR and Western blot) and immunohistochemistry	Therapeutic target	[65]
miRNA-191a-5p	OVX rats	Upregulated	After miR-191 antagonization, miRNA knocks down miR-191a, BDNF expression is upregulated, and PSD-induced cognitive impairment in OVX rats is avoided	miRNA expression level (RT-qPCR) and behavior assay (MWM test)	Therapeutic target	[66]
miR-146a	APP/PS1 transgenic mice	NA	In transgenic mice, overexpression of the microglia-specific miR-146a prevented neuronal loss, attenuated neuroinflammation, decreased Aβ levels, and improved plaque formation	miRNA expression level (RT-qPCR), pro-inflammatory cytokines (ELISA), and behavior assay (MWM test and NOR)	Promising target	[62]
miR-107	AD patients and Aβ-treated SH-SY5Y cells	Downregulated	miR-107 deactivated the FGFR2/PI3K/Akt pathway by inhibiting the expression FGF7	RT-qPCR, luciferase reporter assay, and Western blot	Therapeutic target	[67]
miR-134	Mice	NA	miR-134 could directly bind to BDNF 3′-UTR and was noticeably downregulated by Rb1 in the hippocampus of CUMS-exposed mice	miRNA expression level (RT-qPCR and Western blot) and behavior assay (OFT, TST, FST, and SPT)	Therapeutic target	[68]
miR-134-5p	Rats	Overexpressed	Post-transcriptional control of CREB-1 and BDNF	RT-qPCR and Western blot	Therapeutic target	[69]
miRNA-326	AD mice	NA	Targeting VAV1 in AD, upregulating miR326 reduces tau phosphorylation and neuron death by targeting the JNK signaling cascade	miRNA expression level (RT-qPCR, microarray, and Western blot), quantification of Aβ1–40 and Aβ1–42 (ELISA), and behavior assay (MWM)	Therapeutic target	[70]
MiRNA-199a-5p	MDD patient	Upregulated	WNT2 can be targeted by miR-199a-5p to promote the onset of depression via controlling CREB/BDNF signaling	miRNA expression level (RT-qPCR and Western blot) and behavior assay (OFT, TST, FST, and SPT)	Therapeutic target	[71]
miR-219-5p	AD clinical sample and SH-SY5Y cells	Overexpressed in brain tissues	Inhibits tau phosphorylation by inhibiting TTBK1 and GSK-3β	miRNA expression level	Therapeutic target	[72]
miR-31	3xTg-AD animals and HT-22, HEK293, and SH-SY5Y cells	Downregulated	Enhances cognitive function and eliminates Aβ pathology by focusing on APP and BACE1	miRNA expression level (RT-qPCR and Western blot) and behavior assay	Therapeutic target for AD	[73]
miR-34c	SAMP8 mice and HT-22 cells	Upregulated	Enhanced memory and synaptic impairments caused by miR-34c targeting SYT1 via the ROS-JNK-p53 pathway	miRNA expression level (RT-qPCR and Western blot) and behavior assay (MWM and NOR)	Promising novel therapeutic target	[54]
mir-455-3p	Transgenic mice with miR-455-3p and miR-455-3p KO mice	Downregulated	Improved neural development, lifespan, synapse, cognitive function, and mitochondrial and synaptic functioning in transgenic mice	mRNA expression level (RT-qPCR and Western blot) and behavior assay (MWM)	AD therapy	[74]
miR-298	U373 and HeLa cells	NA	Soluble Aβ levels are decreased as a consequence of miR-298 negative regulation of APP and BACE1 expression	miRNA expression level (RT-qPCR and Western blot), quantification of Aβ1–40 and Aβ1–42 (ELISA), and behavior assay (MWM)	Suitable target for AD therapy	[75]
miR-132	RenCell VM human NPCs, H9 cells, human hippocampal progenitor/stem cell line (HPC0A07/03C), and human hippocampi and serum	Downregulated	AD mice’s neurogenic and memory impairments are restored by overexpression in adult NSCs	miRNA level (reverse transcription and real-time PCR) and behavior assay	Therapeutic potential	[76]
miR-132	Seventy SPF Sprague–Dawley rats treated with 20 μg Aβ25-35	Upregulated	Rats with AD exhibited improved cognitive performance when miR-132 was able to suppress the development of iNOS in the hippocampus and oxidative stress by blocking MAPK1 expression	miRNA expression level (RT-qPCR and Western blot), luciferase reporter assay, and behavior assay (MWM)	Novel clinical target	[77]
miR-132	Postmortem brain samples and neuron cells	Overexpressed	Control cell death and the GTDC-1/CDK-5/Tau phosphorylation signaling pathway	qRT-PCR, Western blot, and immunoprecipitation	Potential therapeutic target	[78]
miR-485-5p	APP/PS1 mice	Downregulated	Targets PACS1 in pericytes, promoting viability and preventing pericyte death	miRNA expression level (RT-qPCR and Western blot) and behavior assay (MWM and FCT)	Suitable target for AD therapeutic	[79]
miR-144-3p	APPswe/PS1dE9 mice	Upregulated	An antagonist of miR-144-3p into the hippocampi partially amended cholinergic degeneration and synaptic/memory impairments by raising tPA protein levels and modifying the proNGF/NGF ratio	miRNA expression level (RT-qPCR, Western blot) and behavior assay (Context–Place Memory Test)	Therapeutic target	[80]
miR155	APP/PSEN1 mice	Downregulated	Linked to an increase in the incidence and size of Aβ plaques in mice aged 4 and 8 months	miRNA expression level (RT-qPCR, Western blot), quantification of Aβ1–40 and Aβ1–42 (ELISA), behavior assay (Barnes maze), and Field electrophysiology	Therapeutic target	[81]
miR-146a-5p	AD clinical sample and AD-HHNs cells	Upregulated	In hippocampus neuronal cells, NF-kB-induced overexpression of miR-146a-5p facilitated oxidative stress and pyroptosis via TIGAR	miRNA expression level (RT-qPCR and Western blot) and quantification of oxidative stress markers (ELISA)	Therapeutic target	[82]
miR-342-3p	3xTg-AD mice and HT22 mouse hippocampal neuronal cells	Upregulated	Elevation of neuronal miR-342-3p in Aβ-challenged HT22 hippocampus neuronal cells is linked to an increase in c-Jun N-terminal kinase activation and, ultimately, neuronal death	miRNA expression level (RT-qPCR and Western blot), quantification of Aβ1–40 and Aβ1–42 (ELISA), and behavior assay (RAWM)	Therapeutic target	[83]
miR-124	Neuroblastoma Neuro-2a cells and P301S transgenic mice	Overexpressed	Aberrant miR-124/PTPN1 signaling in the hippocampus resulted in AD-like tau pathology, which included several sites of hyperphosphorylation, insolubility, and somadendritic aggregation in addition to learning/memory deficits	miRNA expression level (RT-qPCR and Western blot) and behavior assay (fear conditioning test, Morris water maze, and elevated plus maze)	Therapeutic strategy	[84]
miR-124-3p	Human neural cell line, HCN-2, and APP/PS1-AD mice	Downregulated	Considerably improved AD-mouse behavior, and considerably decreased Aβ deposition. Findings point to the critical role that miR-124-3p’s post-transcriptional regulation of calpain plays in the onset of AD	RT-qPCR, Western blot, and social recognition test (SRT)	Therapeutic target	[85]
miR-148a-3p	Twenty-one AD patients, APP/PS1 mice, SAMP8 mice, and senescence-accelerated resistance mice	Downregulated	The direct targeting of the p35/CDK5 and PTEN/p38 MAPK pathways by the upregulation of miR-148a-3p has been observed to protect neuronal cells against Aβ-associated tau hyperphosphorylation	RNA sequencing, qRT-PCR, and Western blot	Potential therapeutic target	[86]
miR-17	5xFAD mice	Overexpressed	Impaired microglial autophagic clearance of Aβ is caused by decreased NBR1 expression and miR-17-mediated autophagy suppression, enhancing autophagy protein expression through the inhibition of increased miR-17-enhanced Aβ breakdown in vitro in microglia	miRNA expression level (RT-qPCR and Western blot)	Therapeutic interventions	[87]
miRNA-15b and miRNA-125b	AD clinical serum sample	Downregulated	A mechanistic connection between miRNA-125b and Aβ-independent neurotoxic pathways, as well as a possible protective anti-Aβ activity of miRNA-15b	miRNA expression level (RT-qPCR) and PET data acquisition	Therapeutic target	[88]
miR-155	C57/BL6 wild-type	Up/downregulated	miR-155 influences the capacity of microglia to catabolize Aβ1-42	miRNA expression level (RT-qPCR)	Therapeutic target	[89]
miRNA-101	AD clinical blood sample, APPswe/PS1DE9 transgenic mice, and SH-SY5Y cells	Downregulated	After miRNA-101a was overexpressed in AD model cells, MAPK1 and beclin-1 were reduced, indicating that miRNA-101a may control the development of autophagy in AD	miRNA expression level (RT-qPCR and Western blot) and dual luciferase assay	Diagnosis	[90]
miR-34a	miR-34a+/transgenic mice	Overexpressed	Rapid onset of AD-like neuropathology and cognitive impairment is caused by the overexpression model	Behavior assay (Y-maze and T-maze)	Therapeutic target	[91]
miRNA-15b and miRNA-125b	AD clinical blood samples		Possible function of miRNAs-15b and -125b as putative miRNA biomarkers of AD pathogenesis	miRNA expression level (RT-qPCR) and PET data acquisition	Biomarkers of AD pathophysiology	[88]
miR-193a-3p	Rat PC12 and SHSY5Y cell lines	Downregulated	miR-193a-3p reduces Aβ-induced neurotoxicity by targeting PTEN	Luciferase reporter assay	Novel biomarker	[92]
miR-212	Plasma from AD patients and Aβ25–35-treated SH-SY5Y and IMR-32 cells	Downregulated	miR-212 controlled PDCD4 in Aβ25–35-treated SH-SY5Y and IMR-32 cells to regulate cell proliferation and death through the PI3K/AKT signaling pathway	qRT-PCR and Western blot	Therapeutic target	[93]
miR-146a and miR-181a	Forty-five patients with MCI	Upregulated	Elevated blood levels of miR-146a and miR-181a in individuals with MCI who experience a progressive loss in their cognitive function, along with a relationship between these changes and markers of the illness and AD risk factors	qRT-PCR and ApoE genotyping	Biomarker	[94]
miR-342-5p	Transgenic APP mice (Exo-APP) or C57BL/6 littermates (Exo-CTL), and individuals with HC (40) and AD (40)	Downregulated	Ameliorates Aβ formation via targeting BACE 1 in AD	miRNA expression level (RT-qPCR and Western blot) and dual luciferase assay	Biomarker	[57]
miR-143-3p	SH-SY5Y cells	Upregulated	miR-143-3p suppression targets NRG1, which enhances neuronal survival; the miR-143-3p/NRG1 axis is a possible target for therapy	miRNA expression level (RT-qPCR and Western blot) and dual luciferase assay	Biomarker	[95]
miR-26b	PC12 cells	Upregulated	In the PC12 cellular AD model, miR-26b causes cell death, downregulates NEP expression, and impairs neurite outgrowth	miRNA expression level (RT-qPCR and Western blot)	Biomarker as well as therapeutic target	[96]
miR-483-5p	HEK293 cells, neuroblastoma SK-N-MC cells, and neonatal human dermal fibroblasts	Upregulated	By controlling ERK1 and ERK2 at the mRNA and protein levels, miR-483-5p lowers tau phosphorylation through direct ERK1/2 repression and reduces the amounts of both kinases’ phosphorylated forms	miRNA expression level (RT-qPCR) and dual luciferase assay	Biomarker	[97]
miR-331-3p	AD patients and Aβ1–40 treated SH-SY5Y cells	Downregulated	miR-331-3p may have a neuroprotective function via controlling the expression of pro-inflammatory cytokines and cell survival in Aβ1–40-treated SH-SY5Y cells	Aβ1–40 and Aβ1–42 (ELISA) and dual luciferase assay	Therapeutic as well as diagnostic target	[98]
miR-433	AD patients and Aβ-treated SH-SY5Y and SK-N-SH cells	Downregulated	In SH-SY5Y and SK-N-SH cells, overexpression of miR-433 may be able to reverse the reduction of neuronal viability caused by Aβ; the results of the luciferase activity assay indicated that miR-433 in neuronal cells targeted the gene JAK2	miRNA expression level (RT-qPCR) and dual luciferase assay	Diagnostic biomarker as well as potential therapeutic target	[99]
miR-4422-5p	HEK-293T cells, SH-SY5Y cells, and the A549 cell line	Downregulated	In HEK293T cells, miR-4422-5p can bind directly to the 30 UTR of the GSAP and BACE1 genes, hence decreasing the function of these genes	Dual luciferase assays, Western blotting, and immunocytochemistry	Potential biomarker and therapeutic target	[100]
miR-200a-3p	APP/PS1 mice	Downregulated	Inhibits the expression of BACE1—miR-200a-3p can either directly or indirectly decrease the overproduction of Aβ; additionally, it can reduce tau hyperphosphorylation by attenuating PKA expression	Dual luciferase assays, Western blotting	Therapeutic target	[101]
miRNA-let-7a-5p	Nine AD patients and in vitro THP-1 cells	Upregulated	Positively regulates TLR3, RIG-I, and MDA5	RT-qPCR assay	Therapeutic target	[102]
miR-10a-5p, miR-142a-5p, miR-146a-5p, miR-155-5p, miR-211-5p, and miR-455-5p	Male APPtg (APPswe/PS1L166P) and TAUtg (THY-Tau22) mice	Deregulated	miRNAs are implicated in a central pathogenic mechanism; however, they do not impair the cognitive function of wild-type mice	miRNA expression level (RT-qPCR and Western blot) and behavior assay (MWM, NOR, CFR, and SPSN)	Prognostic marker	[103]
miR-149-5p	AD patients and healthy volunteers	Upregulated	miR-149–5p influenced the expression of KAT8 and H4K16ac, related to AD pathogenesis	RT-qPCR, Western blot, and dual luciferase assays	Potential drug target	[104]

**Table 2 ijms-25-09936-t002:** A summary of preclinical studies for AD-specific miRNA diagnosis.

miRNA	Patients/Healthy Control (Number of Individuals)/Cell Line/Experimental Animal Model	Sample	Technique	Upregulated/Downregulated	Parameter Study and Technique Used	Concluding Remark	Reference
miR-193a-3p	Sporadic AD patients (108) and controls (93)	Serum	qRT-PCR	Downregulated	The results showed that the diagnostic sensitivity was 89.8% and the specificity was 77.4%	Diagnosis biomarker	[92]
miR-106b	AD patients (106)	Serum	qRT-PCR	Downregulated	qRT-PCR results showed a specificity and sensitivity of around 62% and 94%, respectively	Diagnosis biomarker	[132]
miRNA-193b (exosomal)	APP/PS1 mice, SCD patients (89), MCI patients (92), and DAT patients (92)	CSF and serum	TaqMan qPCR & exosomal RNA ELISA	Upregulated	miR-193b level is higher in the CSF of MCI and DAT patients	Early diagnostic biomarker	[133]
miR-384	SCD patients (45), aMCI patients (50) AD (40), and healthy individuals (30)	Exosomal plasma	RT qPCR and Western blot	Upregulated	The levels of miR-384 were considerably greater in the exosomal plasma of the SCD, aMCI, and AD groups	Diagnosis of SCD	[134]
miR-342-5p	AD patients (19)	Plasma	RT-qPCR	Downregulated	After two years of follow-up, patients with mild AD showed a higher rate of cognitive deterioration in correlation with lower levels of miR-342-5p in plasma	Diagnostic biomarker	[135]
miR-103 and miR-107	AD patients (120), PD patients (120), and controls (120)	Peripheral blood samples	RT-qPCR	Downregulated	For miR-103, the sensitivity and specificity were 80.0% and 84.2%, respectively; for miR-107, they were 77.5% and 59.2%, respectively	Biomarker able to differentiate between disease risk and disease progression stage	[136]
miR-373 and miR-204	AD patients (18), moderate AD patients (18), and cognitively healthy individuals (21)	Plasma	RT-qPCR	Downregulated	Statistically noteworthy reduction in the expression of miR-204 and miR-373 in the mild and moderate AD groups compared to the healthy group	Biomarker	[137]
miR-202	AD patients (121) and controls (86)	Serum	RT-qPCR	Downregulated	Serum miR-202 level could be differentiate between ACI patients and healthy individuals	Potential diagnostic biomarker	[31]
miR-125b	AD patients (105)	Serum	qRT-PCR	Downregulated	The specificity and sensitivity of the assay were 68.3% and 80.8%, respectively	Noninvasive biomarker for AD	[138]
miR-9	AD patients (36) and controls (38)	Serum	qRT-PCR	Downregulated	The groups’ concentrations of miR-9-5p varied, with patients with AD showing a median 3-fold drop in circulating levels when compared to controls	Clinical biomarker	[139]
miR-193b	APP/PS1 mice; MCI and DAT patients	Serum, plasma, and CSF derived exosomes	qRT-PCR and Western blot	Decrease	Patients with DAT and MCI had lower serum and plasma levels of exosomal miR-193b than the control groups	Potential non-invasive marker for MCI and DAT patients	[140]
let-7F-5p, miR-1285, miR-107, miR-103a-5p, miR26b-5p, miR-532-5p, miR-151a-3p, miR-161, let-7d-3p, miR-112, and miR-5010-3p	AD patients (106)	Blood	qRT-PCR and NGS	Six upregulated and six downregulated	An accuracy of 93%, and a specificity and sensitivity of 92% and 95%, respectively	Signature biomarker	[141]
has-let-7d-5p and has-let-7g-5p	AD patients (50) and controls (50)	Plasma	qRT-PCR	Upregulated	A sensitivity of 0.82 and specificity of 0.34 for let-7d-5p; has-let-7g-5p showed 0.79 sensitivity and 0.28 specificity	Diagnostic biomarker	[142]
miR-133b	AD patients (105) and controls (98); SH-SY5Y cells with Aβ25-35	Serum	qRT-PCR	Downregulated	Significantly lessening of the Aβ25–35-induced decrease of cell viability was achieved by overexpressing miR-133b	Potential diagnostic biomarker	[143]
miR-30b-5p, miR-22-3p, and miR-378a-3p	AD patients (8) and controls (8)	Serum	qRT-PCR	Deregulated	miRs were significantly deregulated in AD	Possible biomarker	[144]
has-mir-567	MCI (18) and AD patients (18)	Peripheral blood samples	qRT-PCR	Upregulated	Serum from MCI-AD patients showed a higher fold change than that from controls	Signature marker in MCI diagnosis	[145]
miR-92a-3p, miR-181c-5p, and miR-210-3p	HCs (38), MCI (26), patients with AD dementia (56), and FTD patients (27)	Plasma	qRT-PCR and Western blot	Upregulated	An 89.3% accuracy, 84.6% sensitivity, and 85.71% sensitivity of the assay	Potential plasma biomarkers	[146]
miRNA-101a	AD patients and APPswe/PS1DE9 mice	Plasma	miRNA microarray assay and qRT-PCR	Downregulated	A sensitivity of 0.91 and specificity of 0.73	Diagnostic biomarker	[90]
miR-128	AD patients	Serum samples	Real-time fluorescence quantitative PCR	Upregulated	Statistically significant high levels of miR-128	Clinical diagnostic biomarker	[147]
miR-150-5p	Mild DAT and MCI patients	Postmortem AD hippocampus	qRT-PCR and MRI imaging	Upregulated	miR-150-5p levels were found upregulated in the blood of DAT and MCI patients	Clinical blood-based biomarkers for DAT	[148]
miR-34a, miR-29b, and miR-181c	AD patients (23)	Serum	qRT-PCR	Downregulated	AD key pathological factor—Aβ42, TNF-α, and pTau—levels significantly increased in patients with high diagnostic power and assays showed high sensitivity and specificity for target miRs	Diagnostic marker	[149]

MCI, mild cognitive impairment; DAT, Alzheimer’s type dementia; AD, Alzheimer’s disease; PD, Parkinson’s disease; FTD, frontotemporal dementia; aMCI, amnesic mild cognitive impairment.

## Data Availability

Not applicable.

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
