# Peer review of "Insights into the Role of microRNAs as Clinical Tools for Diagnosis, Prognosis, and as Therapeutic Targets in Alzheimer’s Disease"

_ijms, 2024, doi:10.3390/ijms25189936_

Round 1
Reviewer 1 Report
Comments and Suggestions for Authors
In this review, Puranik and Song offer an examination of the influence of microRNAs (miRNAs) in the context of Alzheimer's disease (AD). The authors discuss the intricate ways in which miRNAs are implicated in the etiology of AD, their prospective value as biomarkers, and their emerging potential in therapeutic strategies. The review underscores the plausible or pivotal role of miRNAs in key neurobiological processes such as synaptic plasticity, the production of amyloid-beta (Aβ) peptides, the aggregation of tau proteins, and the modulation of brain-derived neurotrophic factor (BDNF) signaling pathways, all of which are central to the pathophysiology of AD. The article collects a wealth of research identifying specific miRNAs as candidates for therapeutic intervention and as markers for the early detection and staging of AD. The authors posit that the dysregulation of miRNA expression in both the brain and the periphery could offer a dual utility as diagnostic biomarkers and therapeutic targets in AD. This manuscript adeptly consolidates the existing literature, highlighting the nexus between AD and miRNAs. However, to be considered for publication in the International Journal of Molecular Sciences (IJMS), the following issues require attention:
Major Concerns:
- While Figure 1 is a valuable asset for imparting foundational knowledge, another illustrative figure that delineates the interplay between miRNAs, their targets, and AD is imperative. This visual aid should be bifurcated to showcase both the mechanistic links between miRNAs and AD, including potential therapeutic targets, and the role of miRNAs as biomarkers. Such a diagram would serve as a comprehensive synthesis of the extensive tabular data and expedite the reader's understanding of the core themes.
- The content of Sections 3.1 and 3.2 warrants revision. It is inadvisable to cite other reviews within a review article (line 190-195), except perhaps in the introductory segment. Moreover, these sections lack substantive examples and fail to provide a robust argument. The examples cited merely suggest the potential of miRNAs as biomarkers and the correlation of 127 miRNAs with AD. These sections necessitate significant enhancement and expansion.
- Section 4, titled "Challenges and Future Prospects," demands further development and detail. The assertion that "miRNAs exhibit a high degree of targeting specificity" (line 545) is misleading, considering that one miRNA can target up to hundreds of genes. There has been a dearth of discussion, by other review articles, on the barriers preventing miRNAs from entering the pharmaceutical market, which the authors should address in depth. Additionally, the limitations of using miRNAs as diagnostic or prognostic biomarkers, such as variability among individuals, sampling techniques, and sensitivity, should be discussed in this section, as these topics are heavily featured in the manuscript.
Minor Concerns:
- The two sentences at line 16-18 should be combined to form a single, cohesive statement.
- At line 66-67, please provide a more detailed explanation of the "limitations."
- The presence of an introductory sentence at line 219-210 is somewhat jarring and should be reconsidered.
Author Response
Reviewer 1
Comments and Suggestions for Authors
In this review, Puranik and Song offer an examination of the influence of microRNAs (miRNAs) in the context of Alzheimer's disease (AD). The authors discuss the intricate ways in which miRNAs are implicated in the etiology of AD, their prospective value as biomarkers, and their emerging potential in therapeutic strategies. The review underscores the plausible or pivotal role of miRNAs in key neurobiological processes such as synaptic plasticity, the production of amyloid-beta (Aβ) peptides, the aggregation of tau proteins, and the modulation of brain-derived neurotrophic factor (BDNF) signaling pathways, all of which are central to the pathophysiology of AD. The article collects a wealth of research identifying specific miRNAs as candidates for therapeutic intervention and as markers for the early detection and staging of AD. The authors posit that the dysregulation of miRNA expression in both the brain and the periphery could offer a dual utility as diagnostic biomarkers and therapeutic targets in AD. This manuscript adeptly consolidates the existing literature, highlighting the nexus between AD and miRNAs. However, to be considered for publication in the International Journal of Molecular Sciences (IJMS), the following issues require attention:
Response: Thank you so much for providing your valuable time in reviewing the manuscript. We tried to incorporate all your comments in the revised version of the manuscript.
Major Concerns:
- While Figure 1 is a valuable asset for imparting foundational knowledge, another illustrative figure that delineates the interplay between miRNAs, their targets, and AD is imperative. This visual aid should be bifurcated to showcase both the mechanistic links between miRNAs and AD, including potential therapeutic targets, and the role of miRNAs as biomarkers. Such a diagram would serve as a comprehensive synthesis of the extensive tabular data and expedite the reader's understanding of the core themes.
Response: Thank you for your insightful suggestion. We agree that Figure 1 helps provide foundational knowledge, but an additional figure illustrating the interplay between miRNAs, their targets, and Alzheimer's Disease (AD) would significantly enhance the manuscript. We have added 2 figures in the revised MS.
Figure 1: Diagrammatic representation of a possible role of microRNAs in diagnosis, prognosis, and therapeutics.
Figure 2: miRNA is associated with various pathophysiologies of Alzheimer’s disease including tau, Aβ, BACE1, and BDNF signaling-associated pathology and also related to synaptic dysfunction. miRNA associated with AD pathology could be a potential therapeutic target or biomarker for its diagnosis. (Created with BioRender)
- The content of Sections 3.1 and 3.2 warrants revision. It is inadvisable to cite other reviews within a review article (line 190-195), except perhaps in the introductory segment. Moreover, these sections lack substantive examples and fail to provide a robust argument. The examples cited merely suggest the potential of miRNAs as biomarkers and the correlation of 127 miRNAs with AD. These sections necessitate significant enhancement and expansion.
Response: Response 2: Thank you for your feedback. We agree with your valuable suggestion and section 3.1 and 3.2 are revised accordingly.
3.1 MiRNAs in synapse formation
A molecular connection between miR-502-3p and GABAergic synapse activity in AD mouse hippocampal neuronal cell line was examined by Sharma B et al, 2024. The results of the investigation showed that whilst miR-502-3p suppression raised the amount of GABRα1 protein, miR-502-3p overexpression decreased the level of GABRα1. Overall GABA function was discovered to be enhanced as a result of miR-502-3p overexpression, indicating a negative association between miR-502-3p levels and GABAergic synapse function. The whole research suggested that miRNA, namely miR-502-3p, may be a viable therapeutic target to alter the function of GABAergic synapses in AD [110].
Rodriguez-Ortiz CJ and colleagues investigated that miR-181a is a major negative regulator of the cellular events required for synaptic plasticity and memory and leads to synaptic dysfunction and memory impairments in AD [111].
A recent study by Ge J and the team, 2023 revealed that overexpression of miR-431 in the CA1 region of the hippocampus in AD mice improved their memory deficits and synaptic plasticity but did not affect Aβ levels. miR-431 was identified as targeting Smad4, and knocking down Smad4 mitigated synaptic plasticity and memory impairments in APP/PS1 mice by influencing the expression of synaptic proteins such as SAP102. Additionally, the protective effects of miR-431 were negated by Smad4 overexpression, indicating that miR-431 alleviates synaptic damage partly through suppressing Smad4. These results suggest that targeting the miR-431/Smad4 pathway could be a promising approach for AD treatment [112].
Kumar S and colleagues (2022) investigated changes in cytosolic and synaptosomal miRNAs in postmortem brains of AD patients, healthy brains, Tg2576 mice, Tau transgenic mice, and wild-type mice. The study pinpointed synaptosomal miRNAs that were upregulated with AD progression and highlighted three particularly promising miRNAs—miR-501-3p, miR-502-3p, and miR-877-5p—that influence the function of excitatory and inhibitory synapses in AD [113].
3.2 MicroRNAs and brain-derived neurotrophic factor (BDNF) signaling:
Wu BW and colleagues investigated that miR-10a overexpression inhibits the synapse remodeling in the hippocampal neurons and neuronal cell proliferation and apoptosis through the BDNF-TrkB signaling pathway in a rat model of AD [117].
Wei Li and the team's research verified that miRNA-613 controls BDNF expression in SH-SY5Y cells and APP/PS1 transgenic mice. The study first hypothesized that miR-613 targets the 3'-UTR of BDNF and experimentally confirmed that BDNF mRNA and protein expression levels were markedly decreased in the presence of miR-613; conversely, BDNF expression levels were enhanced when miR-613 was suppressed. This result implies that in the AD model, miR-613 negatively regulates BDNF expression [118]
Zhang J et al, 2018 studied the role of miR-322 (homolog of human miR-424) in the Tg2576 mouse model and found that miR-322 stimulates tau phosphorylation by inhibiting the activation of the BDNF–TrkB receptor; in the absence of miR-322, tau phosphorylation is attenuated and TrkB activity is restored. Concludingly research revealed a unique miRNA-dependent mechanism of BDNF degradation in AD pathogenesis, which may motivate therapeutic approaches based on miRNAs or BDNF [119].
- Section 4, titled "Challenges and Future Prospects," demands further development and detail. The assertion that "miRNAs exhibit a high degree of targeting specificity" (line 545) is misleading, considering that one miRNA can target up to hundreds of genes. There has been a dearth of discussion, by other review articles, on the barriers preventing miRNAs from entering the pharmaceutical market, which the authors should address in depth. Additionally, the limitations of using miRNAs as diagnostic or prognostic biomarkers, such as variability among individuals, sampling techniques, and sensitivity, should be discussed in this section, as these topics are heavily featured in the manuscript.
Response: Thank you for your valuable input. Line 545 is modified and crucial suggestions are added in the revised MS. The following paragraph addresses the challenges of miRNA as a diagnostic marker.
From a biological perspective, there is growing evidence that miRNAs are crucial for regulating many biological processes, such as the formation and normal operation of the central nervous system and miRNAs are shown to reflect the degenerative process taking place in the brain tissue and are differentially expressed in patients compared to controls [164]. Numerous pre-analytical and analytical obstacles face the analysis of miRNAs as a potent diagnostic marker. However, standardizing methods and analyses is necessary to bring them into clinical practice and ensure low costs, high specificity, and sensitivity. Another challenge is finding certain miRNA patterns that can discriminate between different conditions such as separating AD patients from healthy people, determining the disease's severity, and separating AD patients from those with neurological disorders is difficult. However, when a sufficiently large panel of miRNAs is analyzed, distinct patterns of miRNAs may still be found concerning certain disorders, even despite these overlaps. This may improve the utility and specificity of miRNAs as biomarkers. The absence of reference level, analytical standardization, and clinical validation remain significant issues with miRNA as a biomarker.
Minor Concerns:
- The two sentences at line 16-18 should be combined to form a single, cohesive statement.
Response: Sentences combined in revised MS.
- At line 66-67, please provide a more detailed explanation of the "limitations."
Response: The sensitivity of instruments required to measure these proteins in the blood and CSF and due to the peripheral production of Aβ, there is significant overlap between diagnostic groups [20].
- The presence of an introductory sentence at line 219-210 is somewhat jarring and should be reconsidered.
Response: Agree with reviewer. The sentence is modified.
Another therapeutic approach is to introduce miRNA-specific target blockers into cells, which results in target-specific mRNA de-repression and boosts mRNA expression.
Top of Form
Thank you for reviewing our article. We have addressed all the changes and corrections suggested by the reviewer. Please let us know if any further modifications or updates are needed.

Reviewer 2 Report
Comments and Suggestions for Authors
This review manuscript aimed to summarize the role of microRNAs as clinical tools for diagnosis, prognosis, and therapy of Alzheimer’s disease. It provides a comprehensive and up-to-date overview of the issue and highlights the complex roles of microRNAs in AD pathophysiology. As discussed by the authors, this knowledge may eventually help address a critical need in AD management and lead to miRNA use as diagnostic bimarkers and therapeutic targets. Although the manuscript is well-organized and scientifically sound, I do have a few comments and suggestions, which may enhance the quality of the paper:
1. The results of numerous pre-clinical studies on miRNAs are summarized in two fairly extensive tables. However, their formatting is suboptimal and visually not very appealing, which can sometimes make it difficult for the reader to navigate and may even seem discouraging. There is certainly room for improvement. It is also noteworthy that most of the papers presented in the tables are not mentioned or discussed in the text, and vice versa. This makes it somewhat challenging for the reader to follow the topic.
2. In the Introduction chapter, the abbreviation 'ND' instead of 'NDD' is used for neurodegenerative diseases in several places.
3. It would be advisable to use a uniform style for labeling microRNAs, especially in chapter and subsection titles. For example, the titles of chapters 2 and 4 use the abbreviation miRNA, while the title of chapter 3 uses the full form microRNA. The same inconsistency is also found in the individual subsections of chapter 3.
4. Page 2, Chapter 1: Northern blot is mentioned as one of the methods conventionally employed to quantify and identify miRNAs. Strangely, both Tables 1 and 2 mention Western blot instead.
5. Page 3, Chapter 2: Please correct the statement 'RNA sequences that are not transcribed into proteins'.
6. Page 3, Chapter 2: It is stated that miRNA biogenesis can occur via two mechanisms, the canonical and noncanonical pathways. However, the following sentences seem to describe only one of the pathways.
7. Page 4, Figure 1: Part 'a' of the figure seems to show the presence of anti-miRNA strand. Is this correct?
8. Page 18: The whole section 'According to a recent study by Sun et al. (2022) [111] approximately 127 miRNAs may have a regulatory role in BDNF expression. Brains, blood, and CSF from patients with AD show upregulation or downregulation of a broad spectrum of miRs. Twenty-seven differentially dysregulated miRNAs involved in amyloidogenesis, inflammation, tau phosphorylation, apoptosis, synaptogenesis, neurotrophy, neuron disintegration, and activation of cell cycle entrance have thus far been identified' is repeated word by word on page 20, but with a different reference (Smith PY and colleagues [119]). Please correct this shortcoming and remove duplicity.
9. Page 21: Please check the accuracy of whole section about miR-146a, ROCK1 and PTEN. Some statements are rather unclear and confusing. For example, you mentioned that ROCK1 binding to PTEN is necessary for PTEN phosphorylation, which promotes tau dephosphorylation. However, you also stated that tau hyperphosphorylation is reduced by ROCK1 inhibition.
10. Page 26: the section based on the papers by Yoon et al., 2022 and Takousis et al., 2019 is unclear in places and should be improved. Similar also applies to the next section.
11. A paper by Pishbin et al., 2023 is listed twice in References (as numbers 27 and 34).
12. Brief explanations of some terms and abbreviations mentioned in the tables and text (e.g., TRBP, RISC complex, CAPN1, DAPK1, PSP, PTEN, etc.) would aid reader comprehension.
Comments on the Quality of English LanguageOverall, while the manuscript is at a good linguistic and stylistic level, there are a few minor errors, inaccuracies and ambiguities in the text that affect the readability of the paper. Hence, the manuscript would benefit from additional proofreading and minor editing of English language. Some examples:
a) Page 1: It is not clear to me what the term 'leakage of neuroinflammation' means.
b) Page 3, Chapter 2: Please improve the wording of this sentences for better clarity: 'The remaining miRNAs are intergenic, produced independently of a host gene, and regulated by their promoters; approximately half of all currently known miRNAs are intragenic, processed primarily from introns and very few exons of protein-coding genes.'
c) Page 4: A more grammatically correct title for chapter 3 would be 'MicroRNA in AD pathophysiology and as a therapeutic target'.
d) Page 19: Please check the accuracy of the following statement: 'Reduced levels of APP in the brain can be predicted by either upregulating endogenous APP expression or directly delivering miR-101 to the central nervous system.'
e) Page 20: Please correct the following statement: 'This finding points to the possible functions of hsa-miR-143-3p in regulating the diseases associated with AD, such as tau dysfunction and Aβ accumulation'.
f) Page 22, Chapter 4: Please improve the following text: 'As discussed in Section 3, miRNAs are involved in the pathophysiology of AD, and patients with neurodegenerative diseases, such as AD, have significantly different amounts of several noncoding RNAs compared to healthy matched controls, [127], indicating that miRNAs may be useful as diagnostic biomarkers. In contrast to these biomarkers, miRNAs can be measured simply and noninvasively and are sufficiently stable in bodily fluids such as blood and CSF'.
g) Some sentences in the text were left unfinished, for example 'Various in vitro and in vivo studies demonstrating the potential role of miRNAs as biomarkers for AD diagnosis are listed in' on page 22 and 'Microarray miRNA profile studies indicated that miR-200a-3p is downregulated, suggesting that miR-200a-3p may be a possible biomarker of' on page 28.
h) Page 25, Table 2: Please improve the following text: 'serums from MCI-AD patients showed the highest fold change and statistical significance than control'.
g) Page 26: Please improve the following sentences 'Functional enrichment analysis demonstrated that in individuals with AD, pathways linked to the immune response, inflammation, and apoptosis were significantly enriched with elevated pathways.'
h) Page 27: Please correct: 'The oligonucleotide inhibitor miR-384'.
Author Response
Reviewer 2
Comments and Suggestions for Authors
This review manuscript aimed to summarize the role of microRNAs as clinical tools for diagnosis, prognosis, and therapy of Alzheimer’s disease. It provides a comprehensive and up-to-date overview of the issue and highlights the complex roles of microRNAs in AD pathophysiology. As discussed by the authors, this knowledge may eventually help address a critical need in AD management and lead to miRNA use as diagnostic bimarkers and therapeutic targets. Although the manuscript is well-organized and scientifically sound, I do have a few comments and suggestions, which may enhance the quality of the paper:
Response: Thank you so much for providing your valuable time in reviewing the manuscript. We tried to incorporate all your comments in the revised version of the manuscript.
- The results of numerous pre-clinical studies on miRNAs are summarized in two fairly extensive tables. However, their formatting is suboptimal and visually not very appealing, which can sometimes make it difficult for the reader to navigate and may even seem discouraging. There is certainly room for improvement. It is also noteworthy that most of the papers presented in the tables are not mentioned or discussed in the text, and vice versa. This makes it somewhat challenging for the reader to follow the topic.
Response: Thank you for your valuable feedback. We appreciate the reviewer's suggestion to include discussions of the papers cited in the tables within the main text. However, due to space constraints, it is not feasible to incorporate detailed discussions for every case within the main text. Instead, we have summarized the relevant studies in Tables 1 and 2 to provide a comprehensive overview while keeping the main text concise. We believe this approach effectively addresses the reviewer's concerns and maintains the clarity of the manuscript.
All other suggestions have been incorporated in the revised MS.
- In the Introduction chapter, the abbreviation 'ND' instead of 'NDD' is used for neurodegenerative diseases in several places.
Response: Thank you so much for providing your valuable suggestion. The changes have been made accordingly throughout the MS.
- It would be advisable to use a uniform style for labeling microRNAs, especially in chapter and subsection titles. For example, the titles of chapters 2 and 4 use the abbreviation miRNA, while the title of chapter 3 uses the full form microRNA. The same inconsistency is also found in the individual subsections of chapter 3.
Response: Thank you so much for providing your valuable suggestion. The changes have been made accordingly throughout the MS.
- Page 2, Chapter 1: Northern blot is mentioned as one of the methods conventionally employed to quantify and identify miRNAs. Strangely, both Tables 1 and 2 mention Western blot instead.
Response: Thank you for this comment however in Chapter 1 it is mentioned that 'conventionally, Northern blot methods, ………. and deep sequencing have all been used to quantify and identify miRNAs. However, these methods are time-consuming, labor-intensive, and require expensive supplies and equipment as well as skilled individuals for testing.
Neither of the studies mentioned in Tables 1 & 2 used the Northern technique in their experiment however protein expression level influenced by specific miRNA is checked by Western blot.
- Page 3, Chapter 2: Please correct the statement 'RNA sequences that are not transcribed into proteins'.
Response: Thank you for your valuable input. The transcribed word is replaced by translated.
- Page 3, Chapter 2: It is stated that miRNA biogenesis can occur via two mechanisms, the canonical and noncanonical pathways. However, the following sentences seem to describe only one of the pathways.
Response: Thank you for highlighting this point. In this section, only a general mechanism of miRNA biogenesis is explained not any specific.
- Page 4, Figure 1: Part 'a' of the figure seems to show the presence of anti-miRNA strand. Is this correct?
Response: Thank you for your valuable input. Yes. Figure 1a (now 2a) shows normal miRNA function.
- Page 18: The whole section 'According to a recent study by Sun et al. (2022) [111] approximately 127 miRNAs may have a regulatory role in BDNF expression. Brains, blood, and CSF from patients with AD show upregulation or downregulation of a broad spectrum of miRs. Twenty-seven differentially dysregulated miRNAs involved in amyloidogenesis, inflammation, tau phosphorylation, apoptosis, synaptogenesis, neurotrophy, neuron disintegration, and activation of cell cycle entrance have thus far been identified' is repeated word by word on page 20, but with a different reference (Smith PY and colleagues [119]). Please correct this shortcoming and remove duplicity.
Response: Thank you for your valuable suggestion. The duplicate sentences are deleted in the revised MS.
- Page 21: Please check the accuracy of whole section about miR-146a, ROCK1 and PTEN. Some statements are rather unclear and confusing. For example, you mentioned that ROCK1 binding to PTEN is necessary for PTEN phosphorylation, which promotes tau dephosphorylation. However, you also stated that tau hyperphosphorylation is reduced by ROCK1 inhibition.
Response: Thank you for your valuable suggestion. The paragraph is modified and corrected.
MiRNA-146a is found upregulated in the AD patient's brain and miR-146a suppressed the ROCK1 activity. ROCK1 binding to PTEN (Phosphatase and TENsin homolog deleted on chromosome 10) is necessary for PTEN phosphorylation, which promotes tau dephosphorylation. AD is also associated with the reduction in PTEN phosphorylation of PTEN and PTEN-immunoreactive temporal lobe pyramidal neurons. The ROCK1-PTEN signaling pathway in neurons may be regulated by miR-146a upregulation, which could lead to aberrant tau hyperphosphorylation in neurons. The results of Wang G et al, 2016 studies provide credence to the theory that an overexpression of dysregulated microRNA-146a in neurons that negatively controls the translation of the ROCK1 protein. Decreased neural PTEN phosphorylation results from a decrease in neuronal ROCK1 protein, which impairs neuronal tau dephosphorylation [129].
- Page 26: the section based on the papers by Yoon et al., 2022 and Takousis et al., 2019 is unclear in places and should be improved. Similar also applies to the next section.
Yoon et al. recently conducted a meta-analysis of 334 miRNAs associated with AD. Of these, 56-miRs were found to upreguled or downreguled in AD and MCI patients than in healthy controls. the study found that upregulation of miRNA is mainly associated with the pathways linked to the immune response, inflammation, and apoptosis. [152]. Similarly, to find original research comparing the differential expression of miRNAs in AD cases against controls, Takousis P et al. conducted a systematic review. Following a comprehensive assessment, data from 147 separate data sets reporting information on 1121 distinct miRNAs were combined from 107 qualified research. A total of 57 miRNAs were discovered to be differentially expressed at study-wide significance across 461 meta-analyses. It was oberved that several miRs had differential expression in the brain, CSF, and blood-derived specimens [153].
Ines Maldonado-Lasuncion IM and colleagues investigated the serum expression of AD-related miRNAs studies and have further discovered that miR-455-3p, miR-34a-5p, and miR-146a-5p are strong biomarkers for AD, as patients with AD have notably elevated levels of these miRNAs.
Minor corrections in further paragraphs are made in the revised MS.
Line 540-542 is deleted in the revised MS.
- A paper by Pishbin et al., 2023 is listed twice in References (as numbers 27 and 34).
Response: Thank you for pointing out this issue. A correction has been made in the revised MS.
- Brief explanations of some terms and abbreviations mentioned in the tables and text (e.g., TRBP, RISC complex, CAPN1, DAPK1, PSP, PTEN, etc.) would aid reader comprehension.
Response: Thank you for your thoughtful suggestion. Following full forms are added in text TRBP- transactivation response element RNA-binding protein, RISC-RNA-induced silencing complex, DAPK1- Death-associated protein kinase 1, PSP- progressive supranuclear palsy, PTEN- Phosphatase and TENsin homolog deleted on chromosome 10, CAPN1- Calpain 1
Comments on the Quality of English Language
Overall, while the manuscript is at a good linguistic and stylistic level, there are a few minor errors, inaccuracies and ambiguities in the text that affect the readability of the paper. Hence, the manuscript would benefit from additional proofreading and minor editing of English language. Some examples:
- a) Page 1: It is not clear to me what the term 'leakage of neuroinflammation' means.
Response: Thank you for pointing out this issue. Changes have been made in the revised MS. It is leakage of the blood-brain barrier (BBB).
- b) Page 3, Chapter 2: Please improve the wording of this sentence for better clarity: 'The remaining miRNAs are intergenic, produced independently of a host gene, and regulated by their promoters; approximately half of all currently known miRNAs are intragenic, processed primarily from introns and very few exons of protein-coding genes.'
Response: Thank you for your valuable suggestion. Corrections have been made.
The generation of mature miRNAs is initiated by the transcription of genes by RNA polymerase II in the nucleus. The miRNAs are intragenic, processed primarily from introns and very few exons of protein-coding genes.
- c) Page 4: A more grammatically correct title for chapter 3 would be 'MicroRNA in AD pathophysiology and as a therapeutic target'.
Response: Suggested changes have been implemented.
- d) Page 19: Please check the accuracy of the following statement: 'Reduced levels of APP in the brain can be predicted by either upregulating endogenous APP expression or directly delivering miR-101 to the central nervous system.'
Response: Lines 347-348 are modified- MiR-101 target sites reside within the APP 3'-UTR and an experimental study confirmed that miR-101 significantly reduced the expression of APP 3'-UTR [123].
- e) Page 20: Please correct the following statement: 'This finding points to the possible functions of hsa-miR-143-3p in regulating the diseases associated with AD, such as tau dysfunction and Aβ accumulation'.
Response: Lines are modified- This finding points to the possible functions of hsa-miR-143-3p in regulating the proteins associated with AD, such as tau dysfunction and Aβ accumulation.
- f) Page 22, Chapter 4: Please improve the following text: 'As discussed in Section 3, miRNAs are involved in the pathophysiology of AD, and patients with neurodegenerative diseases, such as AD, have significantly different amounts of several noncoding RNAs compared to healthy matched controls, [127], indicating that miRNAs may be useful as diagnostic biomarkers. In contrast to these biomarkers, miRNAs can be measured simply and noninvasively and are sufficiently stable in bodily fluids such as blood and CSF'.
Response: Lines are modified - As discussed in Section 3, miRNAs are involved in the pathophysiology of AD, and patients with neurodegenerative diseases, such as AD, have significantly different amounts of several miRNAs compared to healthy controls, [133], indicating that miRNAs may be useful as diagnostic and prognostic biomarkers. MiRNAs can be measured easily, and noninvasively and are stable in bodily fluids such as blood and CSF [124]. Various in vitro and in vivo studies demonstrating the potential role of miRNAs as biomarkers for AD diagnosis are listed in
- g) Some sentences in the text were left unfinished, for example 'Various in vitro and in vivo studies demonstrating the potential role of miRNAs as biomarkers for AD diagnosis are listed in' on page 22 and 'Microarray miRNA profile studies indicated that miR-200a-3p is downregulated, suggesting that miR-200a-3p may be a possible biomarker of' on page 28.
Response: Lines are completed - Various in vitro and in vivo studies demonstrating the potential role of miRNAs as biomarkers for AD diagnosis are listed in Table 2.
Microarray miRNA profile studies indicated that miR-200a-3p is downregulated, sug-gesting that miR-200a-3p may be a possible biomarker of AD.
- h) Page 25, Table 2: Please improve the following text: 'serums from MCI-AD patients showed the highest fold change and statistical significance than control'.
Response: Lines are modified- serums from MCI-AD patients showed the highest fold change than control
- g) Page 26: Please improve the following sentences 'Functional enrichment analysis demonstrated that in individuals with AD, pathways linked to the immune response, inflammation, and apoptosis were significantly enriched with elevated pathways.'
Response: These ambiguous lines are deleted from the revised MS.
- h) Page 27: Please correct: 'The oligonucleotide inhibitor miR-384'.
Response: Lines are corrected.
Thank you for reviewing our article. We have addressed all the changes and corrections suggested by the reviewer. Please let us know if any further modifications or updates are needed.
